# MV3D-MAE: 2D Pre-trained MAEs are Effective 3D Representation Learners

## Abstract

Deep learning's success relies heavily on the availability of extensive labeled datasets. Compared to 2D data, acquiring 3D data is substantially more expensive and time-consuming. Current multi-modal self-supervised approaches often involve converting 3D data into 2D data for parallel multi-modal training, ignoring the prior knowledge of extensively trained 2D models. Therefore, it is important to find ways to utilize 2D feature priors to facilitate the learning process of 3D models. In this paper, we propose **MV3D-MAE**, a masked autoencoder framework that utilizes a pre-trained 2D MAE model to enhance 3D representation learning. Initially, we convert single 3D point clouds into multi-view depth images. Building on a pre-trained 2D MAE model, we adapt the model for multi-view depth image reconstruction by integrating group attention and incorporating additional attention layers. Then we propose a differentiable 3D reconstruction method named Mv-Swin, which maps the reconstructed results back to 3D objects without using camera poses, thereby learning 3D spatial representations. Thus, MV3D-MAE, through the bidirectional transformation between 2D and 3D data, mitigates the differences between modalities and enhances the network's representational performance by leveraging the prior knowledge in the pre-trained 2D MAE. Our model significantly improves performance on few-shot classification and achieves SOTA results in linear Support Vector Machine classification. It also demonstrated competitive performance in other downstream tasks of classification and segmentation in synthetic and real-world datasets.

## 1 Introduction

3D perception has emerged as a central focus in the field of computer vision research. Inspired by the self-supervised training of Mask Autoencoders (MAEs) (He et al., 2022) for 2D images, researchers are now expanding their efforts to develop an array of self-supervised models for 3D perception tasks (He et al., 2022; Zhang et al., 2023; 2022; Pang et al., 2022; Yu et al., 2022). These models begin with a pre-training phase, utilizing masked point cloud data for reconstruction purposes, before being fine-tuned to excel in various downstream tasks. The potential of these 3D masked models will play a significant role in enhancing our understanding of the 3D world.

Most current 3D masked models, such as PointBert (Yu et al., 2022) and PointMAE (Pang et al., 2022), are inspired by the 2D MAE model. They encode 3D point clouds into a sequence of tokens after patching, followed by random masking. Moreover, these models utilize a transformer-based encoder-decoder model to restore masked point cloud features to complete point clouds. Recently, multi-modal point cloud MAEs like joint-MAE (Guo et al., 2023), and I2P-MAE (Zhang et al., 2023) have emerged (Fig. 1(a)). Joint-MAE converts point clouds into images and conducts joint 2D and 3D training through an MAE structure. On the other hand, I2P-MAE uses converted images as guide masks for directing the masking and reconstruction of point clouds. However, there are limitations to consider for these approaches. Using multi-view 2D images to represent 3D objects inevitably results in 3D geometry information loss. Moreover, the aforementioned methods typically focus solely on the mapping transformation from 3D to 2D, neglecting the alignment process required to reconstruct 3D objects from 2D images. Another kind of approach for multi-modal pre-training resembles PointCLIP V2 (Zhu et al., 2023), incorporating contrastive learning (Fig. 1(b)). The most recent method, MM-Point (Yu & Song, 2024), converts point clouds into multi-view 2D images and 3D point clouds for contrastive learning. The inter-modal contrastive learning across 2D and 3D

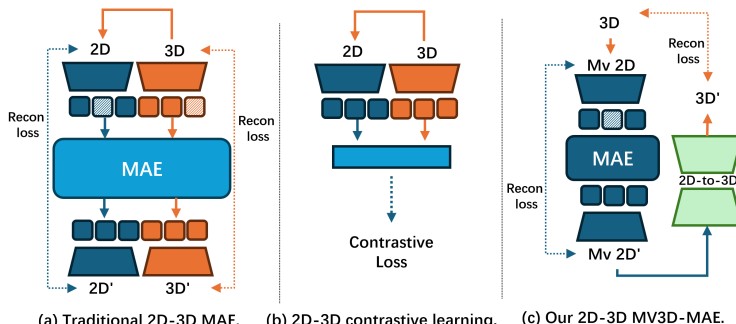

(a) Traditional 2D-3D MAE.  (b) 2D-3D contrastive learning.  (c) Our 2D-3D MV3D-MAE.

Figure 1: **Comparison of 2D-3D MAE, 2D-3D contrastive learning, and MV3D-MAE (Ours).**

domains enriches the feature representation. However, none of the aforementioned methods explicitly execute modality transformation between 2D and 3D, nor do they fully exploit the robust capabilities of existing 2D pre-trained models. Therefore, we raise an evident question: Can a method leverage pre-trained 2D MAE networks, facilitating mutual transformation between 2D and 3D, to achieve self-supervised pre-training for 3D representation?

To tackle this issue, we introduce a novel 2D-3D architecture, **MV3D-MAE** (Fig. 1(c)). First, our model converts 3D point clouds into multi-view 2D depth images. Then we apply a pre-trained 2D MAE to conduct self-supervised learning on 2D depth images. Subsequently, a 2D-to-3D differentiable model reconstructs the depth maps back into 3D point clouds and does supervision. Specifically, we employ a depth rendering model akin to Point-CLIP V2 (Zhu et al., 2023), generating depth maps from ten perspectives of a single point cloud object, each capturing distinct 3D features. Subsequently, these depth maps are processed through masked reconstruction with a ViT-base MAE model, which can be specifically fine-tuned for this task. However, the fundamental 2D MAE can not handle multi-view images directly. Therefore, we introduce group attention to aggregate geometric features from multiple perspectives. Additionally, we incorporated new attention layers into the MAE encoder to adapt the model to the reconstruction of depth images. Traditional methods transform from multi-view 2D depth images to 3D point clouds through discretized geometric mapping, which is non-learnable. Additionally, this mapping imposes strict requirements on both camera poses and data accuracy. To further address the challenge while ensuring the shape features of the point cloud involved in the reconstruction process, we introduce the MV-Swin module. This module is differentiable and merges multi-view depth map features into explicit 3D shapes without requiring camera poses. This capability facilitates end-to-end 2D to 3D learning during pre-training and ensures the alignment between 2D and 3D data. For downstream tasks, we utilize only the MAE encoder, supplemented by a task-specific head. This design not only leverages a pre-trained 2D MAE model as a strong 2D prior for self-supervised 3D representation training but also maximally mitigates the modality differences between 2D and 3D.

Our model is pre-trained on the ShapeNet (Chang et al., 2015) dataset. We then conduct a few downstream tasks, as classification tasks on ModelNet (Wu et al., 2015), ScanObjectNN (Uy et al., 2019) dataset, and segmentation tasks on the ShapeNet Part dataset (Yi et al., 2016) dataset. And we do more experiments in real-world scenarios like 2D-3D retrieval and classification of incomplete datasets. Our main contributions are summarized as follows:

- We leverage a pre-trained 2D MAE model as the 2D prior to enhance the feature learning from multi-view 2D depth images, which are derived from projecting 3D objects. To adapt multi-view depth images to MAE, we introduce group attention after patching images and more attention layers in the MAE encoder, respectively.

- We introduce a novel multi-view reconstruction module, MV-Swin, which enables the transformation of multi-view depth images into point clouds. MV-Swin facilitates the aggregate of multiple depth maps, ensuring a smooth transition from 2D to 3D while aligning features across 2D-3D modalities.

- Through the 2D-3D pre-training with MV3D-MAE, we achieve excellent results in downstream tasks. Notably, in few-shot classification tasks, our model significantly outperforms other 2D-3D self-supervised models, which proves pre-trained 2D MAE is the effective 3D representation learner.

## 2 RELATED WORK

**Self-supervised Point Cloud Learning.** Supervised learning for 3D point clouds has gained traction since the emergence of PointNet (Qi et al., 2017a). However, due to the limited size of 3D point cloud datasets, supervised 3D models have struggled with generalization problems (Uy et al., 2019). To tackle this challenge, self-supervised learning has emerged as a prominent approach for its ability to leverage the vast amounts of unlabeled data effectively (Pang et al., 2022). Self-supervised learning approaches for 3D point clouds can generally be categorized into two classes: one focuses on reconstruction, primarily leveraging masked models, while the other concentrates on discrimination, predominantly employing contrastive learning designs. Although a few 3D models employ contrastive learning for self-supervised learning (Du et al., 2021; Afham et al., 2022), the majority of models still adopt the masked autoencoder (MAE) (He et al., 2022) approach. For example, Point-MAE (Pang et al., 2022), Point-BERT (Yu et al., 2022), and Point-M2AE (Zhang et al., 2022) have marked significant advancements in 3D MAE models. Point-MAE introduces a novel approach by applying the concept of masking and reconstruction to point clouds, aiming to encode the underlying structure of 3D data effectively. On the other hand, Point-BERT adapts the BERT (Devlin et al., 2018) methodology from natural language processing to point clouds scenario, using masked point prediction to capture the complex spatial relationships within the data. Meanwhile, Point-M2AE (Zhang et al., 2022) adopts a multi-scale approach, applying hierarchical masking to point clouds before proceeding with their reconstruction. However, merely learning 3D modality might neglect the implicit semantic and geometric correlations between 2D and 3D data. Compared to these methods, our approach introduces a novel method that embeds 3D spatial features into the 2D self-supervised pre-training process. This strategy innovatively leverages the strengths of both 2D and 3D, offering a comprehensive understanding of the 2D-3D features underlying data.

**2D-3D Representation Learning.** Recent research in self-supervised learning has been taking advantage of the multi-modalities nature of data. These methods primarily involve converting 3D data into 2D data and then applying models to explore the intrinsic relationships between modalities. PiMAE (Chen et al., 2023) introduces a self-supervised pre-training framework based on MAE that fosters interaction between 3D and 2D data to enhance performance in downstream object detection tasks. Joint-MAE (Guo et al., 2023), similar to PiMAE, starts by converting object point clouds into 2D images, followed by multi-modal self-supervised MAE pre-training. I2P-MAE (Zhang et al., 2023) diverges from directly training with 2D images and 3D point clouds together. Instead, it uses 2D images to guide the creation of 3D masks while concurrently reconstructing 2D features. Differing from previous approaches, MM-Point (Yu & Song, 2024) first transforms 3D objects into multi-view images, then employs contrastive learning between these multi-view images and 3D data to learn the similarities between modalities. These methods all leverage 2D data to enhance the performance of 3D tasks, yet they overlook the potential of utilizing already well-developed pre-trained 2D models to improve 3D task outcomes. Moreover, many approaches consider converting 3D data into multi-view 2D data, such as MM-Point (Yu & Song, 2024) and PointCLIP V2 (Zhu et al., 2023), and align the 2D-3D feature with the semantic feature as an implicit way. A valid method is lacking, which can map 2D data back to 3D point clouds to do the explicit alignment in an end-to-end manner. Therefore, in our approach, we propose a method that utilizes the prior 2D information from a pre-trained 2D MAE model to enhance the reconstruction of multi-view images. Then we introduce an Mv-Swin module that maps 2D back into 3D explicitly.

## 3 METHOD

In this section, we elaborate on the pipeline of our proposed method MV3D-MAE (Fig. 2). We describe the transformation of 3D point clouds into multi-view depth maps in Sec. 3.1, the pre-training process using a pre-trained 2D MAE framework in Sec. 3.2, and the MV-Swin architecture for mapping multi-view 2D images back to 3D objects in Sec. 3.3. Additionally, we introduce reconstruction loss for both 2D and 3D data in Sec. 3.4.

### 3.1 FROM 3D TO MULTI-VIEW 2D

The transformation of 3D data into 2D data is a pivotal aspect of our proposed method. Traditional methods often rely on projecting point clouds to depth maps, yet they yield sparse and discontinuous

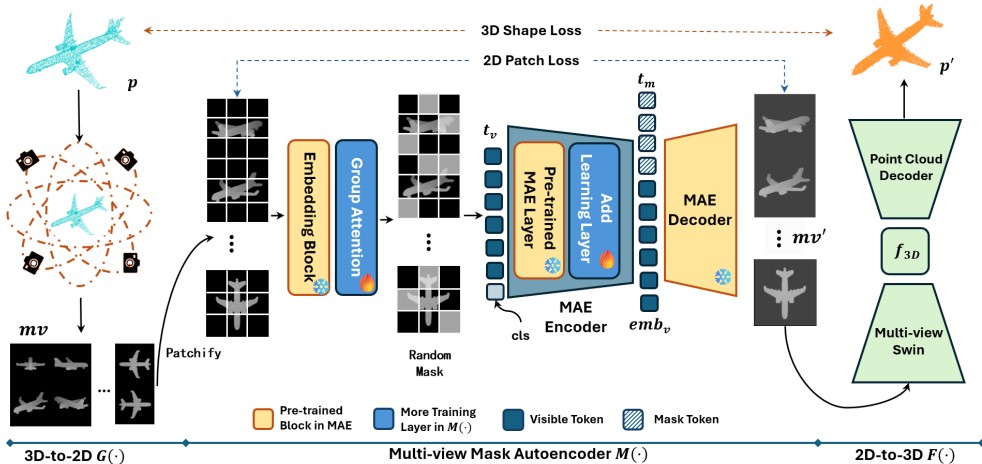

Figure 2: **The pre-training pipeline of MV3D-MAE.** Given a point cloud object $p$, we apply a 3D-to-2D projection function $G(\cdot)$ to obtain multi-view depth images $mv$. Using a pre-trained 2D MAE, we reconstruct these images by dividing them into patches, adding a group attention layer, and performing random masking before passing them through the MAE's self-attention layers. The decoder follows the same structure as the 2D MAE. After reconstructing $mv'$, we use Mv-Swin to transform $mv'$ back into a 3D object $p'$, completing the supervision process.

depth images, hindering subsequent 2D feature learning. Moreover, the discrete nature of their mapping process is incompatible with neural network modeling. Therefore, we seek a continuous mapping function, denoted as $G$, capable of rendering a single 3D object $p$ into multi-view depth images $mv = \{mv_1, mv_2, ..., mv_n\}$. This function facilitates a smoother transition from 3D to 2D data, supporting our objectives without complicating the re-mapping to 3D point clouds.

We adopt the 3D-to-2D rendering pipeline from PointCLIP V2 (Zhu et al., 2023) as our mapping function $G$. This pipeline involves four steps: Voxelize, Densify, Smooth, and Squeeze. Through this process, our pipeline effectively projects a single 3D point cloud object into multi-view depth images with smoothed and continuous shapes, facilitating seamless integration into our framework.

## 3.2 Pre-training with 2D-3D data

**Multi-view consistency.** After acquiring the multi-view depth data, the subsequent step is to reconstruct the multi-view 2D depth images using the MAE framework. Unlike RGB images, multi-view depth images present two distinct characteristics: (1) the need to extract features consistently across multiple views; (2) their single-channel property.

To address these differences, we diverge from the traditional depth images. Firstly, we adjust the depth of images by inverting the background color, opting for a default dark background. Additionally, we replicate the single-channel depth image into three channels, treating it as a conventional three-channel image for further processing.

Furthermore, to ensure consistency across multi-view images, we apply group attention to multi-view images to aggregate features. The group attention mechanism $GroupAttn(\cdot)$ decomposes a tensor of shape $[g \times n, c_{mv}]$ into $[g, n, c_{mv}]$, where $g$ is the number of groups, $n$ is the number of views, $c_{mv}$ is the feature channels, following the self-attention separately to each of the $g$ groups. It's important to note that our group attention operates sequentially within one batch. This approach allows inter-group perspectives to remain unaffected by others, while the intra-group multi-view attention ensures consistency of geometric features across different views.

**Tuning a pre-trained 2D MAE model.** The pre-trained 2D MAE model (He et al., 2022), with its proficiency in understanding features, textures, and patterns specific to 2D images, lays a solid groundwork for feature extraction from depth images. Although depth images primarily focus on spatial depth rather than color information, they possess patterns and structures that the MAE model can skillfully decode. By converting 3D objects into multi-view depth images, employing an MAE to handle these images facilitates the prior knowledge extended into 3D representation learning. This

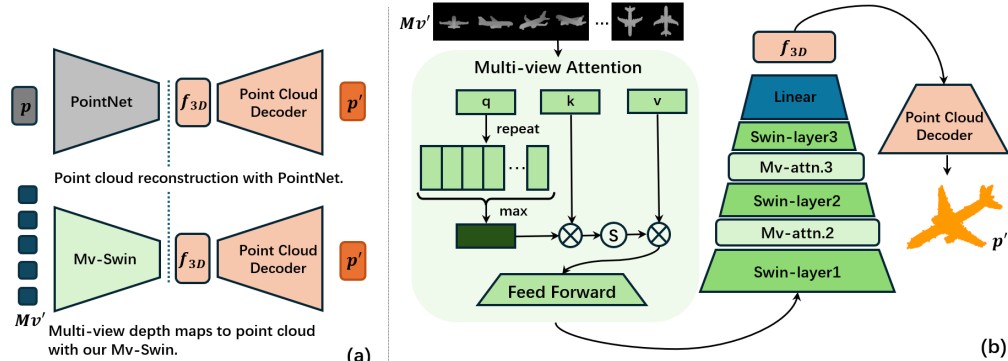

Figure 3: **The overview of Mv-Swin to transform multi-view 2D to 3D.** (a) Inspired by the point cloud reconstruction process, we propose a novel 2D-to-3D reconstruction differentiable process, Mv-Swin. (b) Our model, building upon the existing Swin Transformer block, introduces a Multi-view Attention mechanism to aggregate features from multi-views. Finally, we use a decoder similar to that used in point cloud reconstruction to rebuild the 3D feature $f_{3D}$ back into the point cloud $p'$.

approach not only leverages the strength of 2D insights but also bridges the gap to encompass 3D data understanding, enhancing the model's capacity for interpreting and representing 3D spaces.

To optimize the utilization of feature priors from the pre-trained MAE, we begin by integrating its comprehensive pre-trained components, the embedding layer, encoder, and decoder into our framework. This incorporation, enriched by the addition of a group attention module after the embedding layer, refines the MAE structure for the specific task of reconstructing multi-view depth images. Further, to bolster the encoder's capability in extracting salient features from depth images, we enhance its existing attention mechanisms by introducing extra, trainable multi-head attention layers. Throughout the tuning process, we optimize all layers, drawing upon the 2D priors to significantly refine the MAE model's capability to adeptly navigate the nuances of multi-view depth image reconstruction. This approach ensures that our model leverages the depth of 2D insights to master the challenges of 3D representation learning.

**Multi-view MAE pre-training pipeline.** The whole self-supervise learning process of our multi-view MAE can be described as follows: First, we divide our multi-view depth images *mv* into patches with pre-trained convolution layers, in order to extract the patches' feature as tokens. To accommodate multi-view data, we employ a group attention mechanism $GroupAttn(\cdot)$ to ensure token consistency across different views. Next, we introduce random masking of the 2D tokens with a high ratio, *e.g.*, 70%. This strategy challenges the model to reconstruct the missing information solely based on a limited set of visible tokens, denoted as $t_v$. Consistent with MAE's design, we append the *cls* token to $t_v$ and then feed them input into the MAE encoder. Inside the encoder, we initially utilize pre-trained MAE encoder's multi-head attention layers, denoted as $\mathcal{M}(\cdot)$, to leverage the 2D prior knowledge acquired by the MAE. Subsequently, we introduce a new set of multi-head attention layers, represented as $\mathcal{A}(\cdot)$. These layers are specifically designed to adapt the representation learning process according to the depth images. The initial layers, $\mathcal{M}(\cdot)$, serve as the foundation by providing a comprehensive understanding of image features, which is essential for establishing a robust feature representation baseline. Following this, the introduction of $\mathcal{A}(\cdot)$ layers specifically addresses the nuances of depth images, such as the importance of spatial relationships and depth value variations across different image regions. The embedding $emb_v$ can be formulated as:

$$emb_v = \mathcal{A}(\mathcal{M}(Concat(t_v, cls))). \tag{1}$$

Before feeding the token into the MAE decoder *Dec*, we concatenate the visible tokens $emb_v$, with the masked tokens $T_m$. We utilize the pre-trained 2D MAE decoder to reconstruct the combined token, aiming to reconstruct the multi-view depth images:

$$mv' = Dec(Concat(emb_v, t_m)). \tag{2}$$

### 3.3 FROM MULTI-VIEW 2D TO 3D

Several previous methods have processed 3D point cloud objects by transforming them into multi-view 2D images, including MM-Point (Yu & Song, 2024), PointCLIP V2 (Zhu et al., 2023), and

multi-view point completion model (Hu et al., 2020). These approaches initially convert point clouds into 2D depth images or RGB images and then extract their features to represent individual 3D objects. However, intuitively, the transformation from 3D to multi-view 2D inherently leads to the loss of geometry information of the 3D objects. This loss of information poses challenges, such as the inability of 2D features to directly reconstruct 3D shapes in PointCLIP V2 (Zhu et al., 2023). Moreover, in the case of MV-Completion (Hu et al., 2020), the method relies on traditional discrete mapping algorithms from depth maps to point clouds, often requiring downsampling to achieve point cloud reconstruction. As a result, we propose a novel differentiable algorithm for mapping multi-view depth features to 3D point clouds, MV-Swin, which reconstructs 3D shapes without the need for camera poses.

We revisit the encoder-decoder architecture for the point cloud reconstruction process, taking PointNet (Qi et al., 2017a) as an example for the encoder (Fig. 3a). The encoder initially maps discrete point cloud data into a high-dimensional space, producing a continuous high-dimensional feature representation. The decoder $D$ as Pcl2pcl (Chen et al., 2019), which consists of MLPs, then expands the high-dimensional features to map the features back into the discrete space of the point cloud. Similarly, we can utilize a mapping function $F$ to project multi-view depth images $mv'$ into the high-dimensional feature space. By using the same decoder $D$ as the reconstruction decoder, we can map continuous high-dimensional features $f_{3D}$ back into the discrete space of the point cloud $p'$, thus achieving a differentiable mapping from 2D to 3D: $p' = D(f_{3D}) = D(F(mv'))$.

In our method, we introduce a new module, Mv-Swin (Fig. 3b), to serve as the mapping function $F$. The Swin Transformer (Liu et al., 2021) is adept at aggregating information from multi-scale images, making it particularly suitable for perceiving edge information of depth images to synthesize 3D shapes. However, to integrate multi-view depth image information, a capability lacking in Swin, we insert an Mv-attention module before each Swin module. The Mv-attention module decomposes the original image into patches following the Swin Transformer. Subsequently, patch features are used to calculate query, key, and value. The query features are duplicated according to the number of views, and then the maximum value across multiple views is computed to represent the overall feature of all views, while key and value features remain unchanged. These features are processed by Mv-attention before being fed into the Swin module for further attention processing. The process of Mv-attention and Swin module cascades three times sequentially. Following this, we apply a linear transformation layer, to fuse the multi-view features into a high-dimensional feature $f_{3D}$.

Mv-Swin effectively aggregates multi-view depth features into a unified representation, enabling the conversion of 2D features into a 3D shape. Employing self-supervised training on 2D depth images with a 2D MAE, we can utilize Mv-Swin to convert reconstructed depth maps back into 3D data, facilitating end-to-end learning and 2D-3D alignment. To make the MAE training more efficient, we pre-train the Mv-Swin module initially. During MV3D-MAE training, we freeze Mv-Swin parameters while retraining the point cloud decoder.

### 3.4  2D & 3D Reconstruction

As illustrated in Fig. 2, our loss function emphasizes the reconstruction of both 2D images and 3D shapes. For the 2D reconstruction loss, we employ the Mean Squared Error (MSE) loss, which is consistent with the loss function utilized in MAE. For the reconstruction of 3D shapes, we utilize the traditional Chamfer Distance (CD Loss). Our loss function can thus be represented as follows:

$$\begin{aligned} \mathcal{L}_{2D}(mv, mv') &= MSE(mv, M(mv)) \\ \mathcal{L}_{3D}(p, p') &= CD(p, F(mv')). \end{aligned} \tag{3}$$

Our 2D-3D reconstruction process is sequential, indicating that if the reconstruction of 2D data achieves the desired quality, the reconstruction of 3D data will also be enhanced. Therefore, the overall loss for the MV3D-MAE pre-training is formulated below to reflect this dependency, as $\mathcal{L} = \mathcal{L}_{2D} + \lambda \cdot \mathcal{L}_{3D}$, where $\lambda$ are weight factors. These two self-supervised losses are crucial in enabling our MV3D-MAE structure to effectively learn the representational features of both 2D and 3D data. By tuning the model with the loss, we encourage it to capture the intricate details and complexities inherent in multi-view depth images and corresponding 3D objects. And the classification and segmentation heads' details are explained in the Appendix A.7.

Table 1: **Linear SVM classification results on ModelNet40, ModelNet10 and ScanObjectNN.** *Self.* and *Sup.* represent pre-training with self-supervised and supervised methods.

| Type | Method | Accuracy (%) | | |
|------|--------|------------|-----------|-------------|
| | | ModelNet40 | ModelNet10 | ScanObjectNN |
| *Sup.* | PointNet (Qi et al., 2017a) | 89.2 | - | - |
| | GIFT (Bai et al., 2016) | 89.5 | 91.5 | - |
| | MVCNN (Su et al., 2015) | 89.7 | - | - |
| *Self.* | Point-BERT (Yu et al., 2022) | 87.4 | - | 83.1 |
| | Point-MAE (Zhang et al., 2022) | 91.0 | - | 85.2 |
| | Joint-MAE (Guo et al., 2023) | 92.4 | - | - |
| | CrossPoint (Afham et al., 2022) | 91.2 | - | 81.7 |
| | MM-Point (Yu & Song, 2024) | 92.4 | 95.4 | 87.8 |
| | I2P-MAE (Zhang et al., 2023) | 93.4 | - | 87.1 |
| | **MV3D-MAE** | **93.8** | **95.7** | **88.0** |

## 4 EXPERIMENTS

### 4.1 EXPERIMENTS SETUP

**Pre-training dataset.** Following the precedent setting by prior studies such as (Yu et al., 2022; Zhang et al., 2023; Guo et al., 2023), we select ShapeNet (Chang et al., 2015) as our pre-training dataset. The dataset contains more than 57,000 synthetic 3D shapes from 55 categories. In our pre-training process, we random sample 8192 points from each object as the point cloud input. As the multi-view depth images transform from the 3D objects, we set one object to have 10 multi-view depth images to describe the shape representation and each depth image size $H \times W$ as 224×224 as PointCLIP V2 (Zhu et al., 2023).

**Pre-training settings.** Within the MAE framework, we base our model on the pre-trained 2D MAE model with ViT-base/16 (Dosovitskiy et al., 2020). This fundamental model includes 12 layers in the encoder and 8 layers in the decoder. Building upon this, we add an additional 4 attention layers to the encoder to better adapt the learning of depth representations. For the 2D-to-3D reconstruction module, we follow the Swin Transformer (Liu et al., 2021), setting the window size to 14.

During the pre-training process, our initial step involves pre-training the Mv-Swin module and then freezing its parameters. The mask ratio is set to 0.75. Subsequently, we fully tune the pre-trained 2D MAE structure to adapt the MAE model for the reconstruction of our multi-view depth images. Due to the significant difference in order of magnitude between the 3D CD loss and the 2D MSE loss in our pipeline, we need to apply a larger weight on the 3D loss than the 2D loss to ensure a balance between them. Therefore, we assign a weight of 25 to $\lambda$. The optimization is carried out using the AdamW optimizer, with a learning rate of 1e-4, weight decay of 5e-2, and employing a cosine scheduler for adjusting the learning rate. The batch size is configured at 16, with the training over 300 epochs on four 40GB Nvidia A100 GPUs.

**Downstream tasks.** We choose the classification and part segmentation tasks to validate the performance and generalizability of our pre-trained MV3D-MAE: **(1) Shape classification:** Experiments are conducted on three datasets: ModelNet40 (Wu et al., 2015), ModelNet10 (Wu et al., 2015), and ScanObjectNN (Uy et al., 2019). ModelNet is a widely utilized synthetic dataset for 3D objects. ModelNet40 comprises objects from 40 categories, featuring 9,843 objects for training and 2,468 objects for testing. On the other hand, ModelNet10 consists of 4,899 CAD models from 10 categories. ScanObjectNN is a real-world scanned dataset that includes approximately 2,880 unique point cloud objects, offering a more realistic and challenging setting for evaluating models. **(2) Shape part segmentation:** The ShapeNet Part Dataset (Yi et al., 2016), includes 16,880 models spanning 16 shape categories, with each model consisting of 2 to 6 parts. Adhering to the established evaluation methodology outlined (Qi et al., 2017b), we sample 2,048 points from each shape for analysis. Our evaluation metrics were calculated for each class (classified as cls. mIoU) and the mean IoU was computed across all test instances (denoted as ins. mIoU).

### 4.2 PERFORMANCE ON DOWNSTREAM TASKS

In this section, we evaluate the classification and segmentation performance of our model. Initially, following the latest multi-view self-supervised model, MM-Point (Yu & Song, 2024), we assess the linear Support Vector Machine (SVM) classification capabilities of our model using a pre-trained encoder. Then we evaluate the accuracy of the few-shot classification capability of our model on

ModelNet40. We test classification performance on both the ModelNet40 and ScanObjectNN datasets. Finally, we test the segmentation capability of our model on ShapeNet Part. And also illustrate the 2D-3D retrieval applications.

**Linear SVM Classification.** As indicated in Tab. 1, MV3D-MAE outperforms the latest multi-view self-supervised learning framework, MM-Point, across all three datasets. Our model achieved performance levels of 93.8%, 96.7%, and 88.0%, respectively. Furthermore, compared to other methods like I2P-MAE, which utilize 2D features, our approach delivers superior results, showcasing significant performance enhancements. The consistent performance exhibited by our model on both synthetic and real-world datasets further solidifies its efficacy.

**Few-shot Classification.** Few-shot learning is crucial in scenarios where only limited labeled data is available, mimicking real-world scenarios where acquiring large labeled datasets can be costly and time-consuming. Following the experimental setting of the previous studies (Pang et al., 2022; Guo et al., 2023; Yu & Song, 2024), we conduct few-shot classification on ModelNet40 using the "K-way, N-shot" settings, repeated 10 times. This involves randomly selecting N

Table 2: **Few-shot classification on ModelNet40.** The average accuracy (%) and standard deviation (%) from 10 independent experiments.

| Method | 5-way | | 10-way | |
|---|---|---|---|---|
| | 10-shot | 20-shot | 10-shot | 20-shot |
| PointNet (Qi et al., 2017a) | 52.0 ± 3.8 | 57.8 ± 4.9 | 46.6 ± 4.3 | 35.2 ± 4.8 |
| PointNet + CrossPoint (Afham et al., 2022) | 90.9 ± 4.8 | 93.5 ± 4.4 | 84.6 ± 4.7 | 90.2 ± 2.2 |
| DGCNN (Wang et al., 2019) | 31.6 ± 2.8 | 40.8 ± 4.6 | 19.9 ± 2.1 | 16.9 ± 1.5 |
| DGCNN + CrossPoint (Afham et al., 2022) | 92.5 ± 3.0 | 94.9 ± 2.1 | 83.6 ± 5.3 | 87.9 ± 4.2 |
| Transformer (Vaswani et al., 2017) | 87.8 ± 5.2 | 93.3 ± 4.3 | 84.6 ± 5.5 | 89.4 ± 6.3 |
| Transformer + OcCo (Wang et al., 2021) | 94.0 ± 3.6 | 95.9 ± 2.3 | 89.4 ± 5.1 | 92.4 ± 4.6 |
| Point-BERT (Yu et al., 2022) | 94.6 ± 3.1 | 96.3 ± 2.7 | 91.0 ± 5.4 | 92.7 ± 5.1 |
| MaskPoint (Liu et al., 2022) | 95.0 ± 3.7 | 97.2 ± 1.7 | 91.4 ± 4.0 | 93.4 ± 3.5 |
| Point-MAE (Pang et al., 2022) | 96.3 ± 2.5 | 97.8 ± 1.8 | 92.6 ± 4.1 | 95.0 ± 3.0 |
| MM-Point (Yu & Song, 2024) | 96.5 ± 2.8 | 97.2 ± 1.4 | 90.3 ± 2.1 | 94.1 ± 1.9 |
| Joint-MAE (Guo et al., 2023) | 96.7 ± 2.2 | 97.9 ± 1.8 | 92.6 ± 3.7 | 95.1 ± 2.6 |
| **MV3D-MAE** | **98.0 ± 1.2** | **98.1 ± 1.5** | **93.9 ± 1.8** | **96.2 ± 2.9** |

classes from ModelNet40 and sampling K objects from each selected class. We carried out tests under the configurations of 5-way 10-shot, 5-way 20-shot, 10-way 10-shot, and 10-way 20-shot. As indicated in Tab. 2, we report the mean accuracy and standard deviation for each of these settings. Notably, our approach achieved SOTA performances in every tested setting, and lower standard deviation demonstrating higher stability as well. We attribute this success to the prior knowledge embedded within the pre-trained 2D MAE, which significantly bolstered the model's few-shot learning capabilities. This experiment further facilitates that the model effectively aligns 2D and 3D features.

**Classification.** Subsequently, we evaluated our model's classification performance on ModelNet40 and ScanObjectNN datasets. As shown on the left side in Tab. 3, our model achieved a competitive level of overall classification performance on ModelNet40, reaching a classification accuracy of 94.1%, which is comparable to the performance of joint-MAE. On the ScanObjectNN dataset, our model reaches the SOTA result of 89.85 %. This demonstrates the effectiveness of modifying and fine-tuning a 2D MAE model, enabling it to attain competitive performance on 3D tasks. It indicates that 3D features can be effectively represented through 2D images after being bridged by MV3D-MAE.

**Part Segmentation.** Additionally, to assess the model's effectiveness on fine-grained dense 3D data, we followed prior work and tested our model's segmentation capabilities on the ShapeNet Part dataset. The right side of Tab. 3 showcases a comparison of the segmentation performance between MV3D-MAE and previous models. As indicated in the table, our model achieved competitive results compared to other models, both in terms of class-wise Intersection over cls. mIoU and ins. mIoU. This demonstrates that our approach can effectively leverage 2D data prior knowledge to handle complex 3D tasks.

## 4.3 2D-3D RETRIEVAL APPLICATION

While previous works like Joint-MAE, I2P-MAE, and MM-point have not explored similar retrieval tasks, we think investigating the retrieval capabilities of 2D-3D models is both interesting and feasible. Our model, which leverages a pre-trained 2D MAE, can effectively encode features from RGB images. For the retrieval task, we encode a grayscale-processed RGB image using the MV3D-MAE encoder to obtain image features. The retrieval candidates' features are encoded parts of the multi-view images of 3D objects within the MV3D-MAE structure. We compute the cosine similarity between the query image and the MV3D features, identifying the top 5 most similar features and their corresponding 3D objects from the depth maps.

Table 3: **Comparison on downstream tasks.** The left table shows the accuracy of classification on ModelNet40 and ScanObjectNN dataset. The right table shows the performance of segmentation on ShapeNet Part with cls. mIoU and ins. mIoU.

| Method | Classification | | Segmentation | |
|---|---|---|---|---|
| | ModelNet40 | ScanObjectNN | ins. mIoU | cls. mIoU |
| PointNet (Qi et al., 2017a) | 89.2 | 79.2 | 83.70 | 80.39 |
| PointNet++ (Qi et al., 2017b) | 90.5 | 84.3 | 85.10 | 81.85 |
| DGCNN (Wang et al., 2019) | 92.9 | 86.2 | 85.20 | 82.33 |
| MVNet (Yan et al., 2023) | - | 89.7 | 86.10 | 84.80 |
| Transformer (Vaswani et al., 2017) | - | 80.55 | 85.10 | 83.42 |
| Transformer + OcCo (Wang et al., 2021) | 92.1 | 85.54 | 85.10 | 83.42 |
| Point-BERT (Yu et al., 2022) | 93.2 | 88.12 | 85.60 | 84.11 |
| MaskPoint (Liu et al., 2022) | 93.8 | 88.10 | 86.00 | 84.40 |
| Point-MAE (Pang et al., 2022) | 93.8 | 88.29 | 86.10 | - |
| MM-Point (Yu & Song, 2024) | - | - | - | **85.71** |
| Joint-MAE (Guo et al., 2023) | 93.8 | 88.86 | **86.28** | 85.41 |
| **MV3D-MAE** | **94.1** | **89.85** | 86.24 | 85.48 |

As shown in Figure 5, we test several samples, such as lamp, helmet, microphone, and guitar. These items could all retrieve corresponding or similar objects, which means that although our model encodes depth structures, it effectively captures the shape structure of objects. Using grayscale processed RGB images, it can also find closely related 2D depth maps.

## 4.4 ABLATION STUDIES

In our ablation study, we explore the influence of different components within Mv-MAE on the classification task performance on ModelNet40. To quantitatively assess this, we select the Chamfer Distance (Eq. (3)) as the metric for reconstruction quality. More ablation studies are shown in Appendix A.6 about the impact of various components within the Mv-Swin on the multi-view reconstruction performance.

**Real-world Generalization.** In the real world, researchers often find it challenging to obtain complete scans or all viewpoints of a 3D object. Existing 3D object point cloud perception pipelines rarely experiment with or focus on this aspect. To address this problem, we include two experiments to demonstrate the stability of MV3D-MAE. Based on pre-trained depth images from 10 viewpoints, we randomly remove 2, 5, 7, and in the most extreme case, 9 depth maps in the downstream classification task. We apply the Gaussian noise as a substitute input to the model to verify the reliability of the pre-trained multi-view base model's performance when viewpoints are missing in downstream tasks and to conduct an analysis. As shown in Fig. 5, the performance of the fine-tuned model gradually decreases as 2 to 9 views are removed. Removing 2 views had almost no impact on performance, proving the model's robustness. The most significant drop occurred from 3 views to 1 view, with accuracy decreasing to 89.6%. When generalized to incomplete views, performance dropped significantly with only one view (71%), while 5 views maintained 93%. This shows the model's stability with up to 5 missing views.

Also, we test MV3D-MAE's performance on incomplete point cloud datasets, which are common in real-world scenarios. We test our model on the PCN Dataset and create an incomplete ModelNet40-Crop dataset by cropping ModelNet40 to 4096 points. We compare our model's fine-tuned performance on PCN and

Table 4: **Performance on incomplete datasets.**

| Dataset | Our Model (%) | Point-MAE (%) |
|---|---|---|
| PCN dataset | 92.2 | 91.1 |
| MN40-Crop (Fine-tuned) | 92.0 | 90.4 |
| MN40-Crop (Generalized) | 67.3 | 69.4 |

ModelNet40-Crop with the classical point cloud self-supervised model, Point-MAE. We sample 8000 data points from the PCN dataset for training and use 1200 data points for testing. As shown in Tab. 4, our method achieves an accuracy of 92.2% on PCN, showing credible results even with incomplete point cloud structures. Point-MAE achieves 91.1%. On the ModelNet40-Crop dataset, our model maintains stable classification performance with an accuracy of 92.04%, while Point-MAE achieves 90.41%. When transferring the model fine-tuned on ModelNet40, our model maintains an accuracy of 67.3%, and Point-MAE achieves 69.4%. This discrepancy is within the expected range because our training is based on depth maps, and there is a certain domain gap between the depth maps generated by incomplete point clouds and those from complete point clouds.

Figure 4: **The retrieved examples across four different classes.** Using 2D RGB images, we first retrieve similar depth maps and then find the corresponding 3D objects.

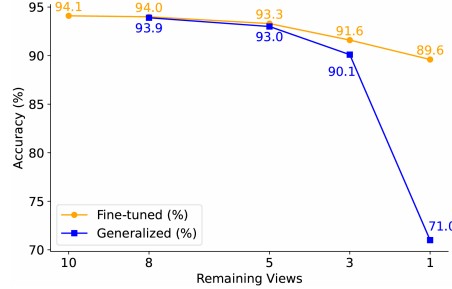

Figure 5: **Classification performance *v.s.* the remaining views.** We test fine-tuned and generalized ways for classification with different remaining views.

**Multi-view MAE Design.** Building upon the foundation of the basic MAE model, we make enhancements by incorporating group attention and increasing the number of attention layers in the encoder. In this section, we conduct ablation experiments using unpretrained MAE (ViT-base) and MAE variants that either lack the added group attention or include the additional layers, to evaluate the impact of these modifications. Tab. 5 shows that using an MAE model without 2D pre-training results in a classification performance of only 90.8%. In contrast, a pre-trained MAE model boosts performance to 93.0% directly. This demonstrates that the prior knowledge in a pre-trained MAE, when subjected to our multi-view training, can better understand 3D spatial structures. Furthermore, the inclusion of group attention and the addition of more attention layers further enhance the model's learning capability for multi-view models, which enhances the performance to 94.1%.

**The Weight Balance of $\lambda$.** To verify the impact of our added 3D module on the 2D experiments, we need to reset the value of $\lambda$ to test the experimental results on the ModelNet40 classification task. Here, we adjust the value of $\lambda$ to 0, 25, 50, and 100 for the experiments Tab. 5. When we set $\lambda$ to 0, it means that the model is pre-trained using only the 2D MAE part without using 3D supervision. At this point, while the

Table 5: **Ablation study on ModelNet40 classification.**

| Pre-trained | Add Layers | Group Attn. | $\lambda$ | Accuracy (%) |
|:---:|:---:|:---:|:---:|:---:|
| - | - | - | 25 | 90.8 |
| ✓ | - | - | 25 | 93.0 |
| ✓ | ✓ | - | 25 | 93.5 |
| ✓ | ✓ | ✓ | 0 | 91.4 |
| ✓ | ✓ | ✓ | 25 | **94.1** |
| ✓ | ✓ | ✓ | 50 | 93.1 |
| ✓ | ✓ | ✓ | 100 | 92.2 |

model can achieve certain results in downstream classification tasks, it lacks a better ability to distinguish 3D objects. On the other hand, if $\lambda$ is set too high, the 3D supervision can affect the 2D MAE's reconstruction ability to some extent, thereby impacting the performance of the downstream model. When we set $\lambda$ as 25, the model achieves the best performance on the downstream classification task.

## 5 CONCLUSION

In this work, we present MV3D-MAE, a novel self-supervised training framework that enriches 3D pre-training with 2D priors within the pre-trained 2D MAE. Diverging from traditional 2D-to-3D pre-training methods, our strategy initiates by converting individual 3D objects into multi-view depth images. Leveraging a pre-trained 2D MAE for prior knowledge, we introduce an innovative multi-view MAE structure tailored for the self-supervised pre-training of multi-view depth images. Following the reconstruction of 2D depth images, our MV-Swin module facilitates the transition back to 3D point clouds, enabling effective 3D reconstruction. Significantly, the group attention mechanism within our multi-view MAE structure captures consistency across multiple views, while the integration of new self-attention layers allows the adaptation of the pre-trained MAE for depth image feature learning. The MV-Swin module further ensures semantic feature alignment from 2D to 3D, exemplifying our method's capacity to merge 2D insight with 3D representation learning seamlessly. Our model not only achieves state-of-the-art results in linear SVM and few-shot classification tasks but also exhibits competitive results in other evaluations. This highlights that leveraging the pre-existing 2D knowledge can elevate the expressive quality of 3D information and enhance the effectiveness of subsequent 3D endeavors.

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

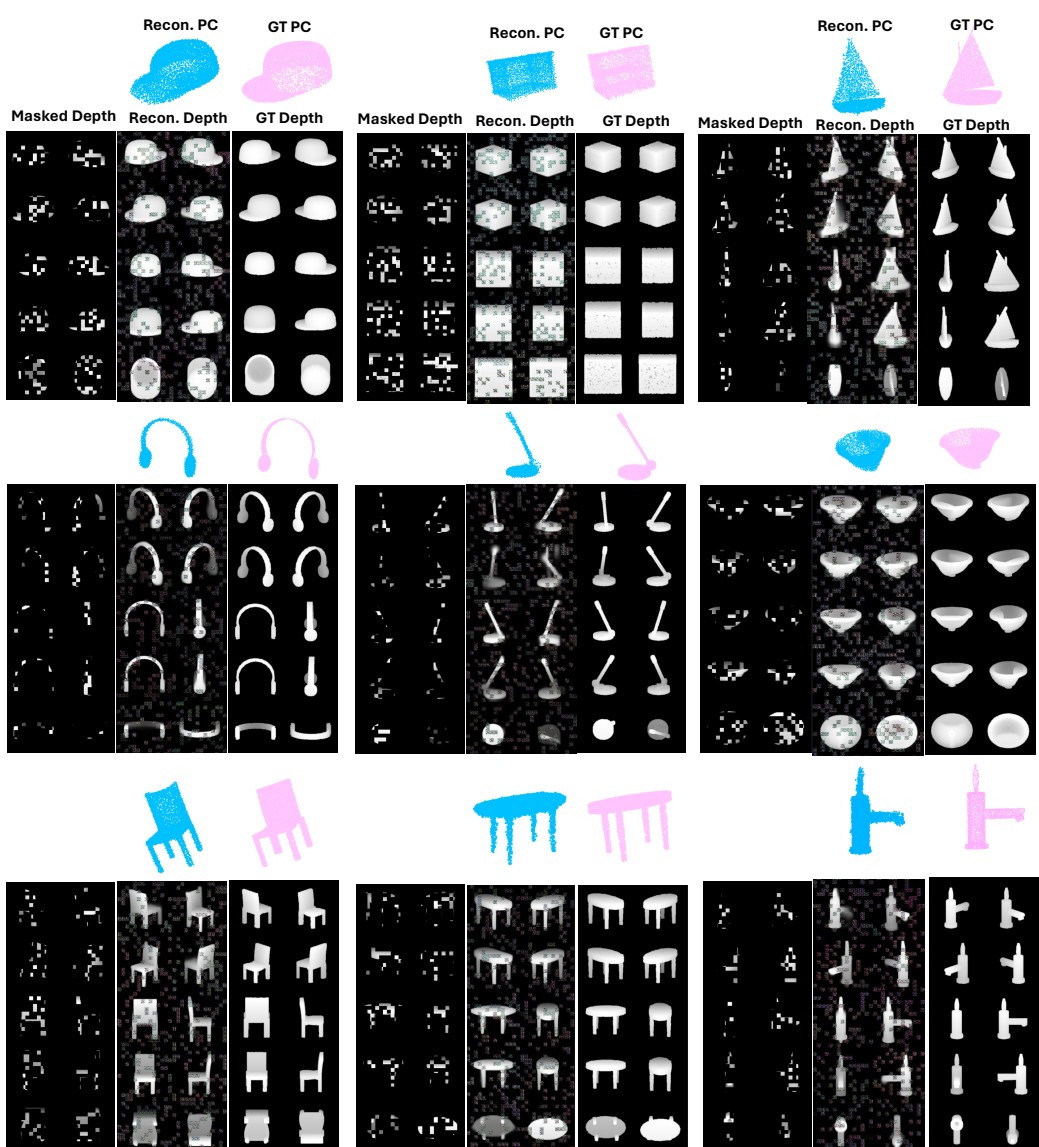

Figure 6: **The visualization of the 2D and 3D reconstruction on *validation* dataset.**[†] We illustrate the ground truth of the 3D objects and multi-view depth maps, the masked multi-view depth maps with 75% mask ratio, and reconstructed depth maps. Please zoom in to see the details. [†]*Following MAE He et al. (2022), there is no loss computed on visible patches, the model output on visible patches is qualitatively worse. By overlaying the output onto the visible patches, one could enhance the visual quality. However, we deliberately choose not to employ this, allowing us to showcase the behaviour of our method.*

## A  APPENDIX

### A.1  VISUALIZATION OF THE 2D&3D RECONSTRUCTION

We showcase the visualization results on the 2D and 3D reconstruction on the validation dataset in the Fig. 6 to display MV3D-MAE's 3D and 2D representation learning capability and modality alignment. The Fig. 6 shows that even with the high mask ratio (75%), MV3D-MAE still can reconstruct the high-quality 2D and 3D objects, which can lead to enhanced performance on downstream tasks.

## A.2 Details of the 3D-to-2D Projection

In 3D-to-2D projector $G(\cdot)$, we utilize a similar projection method as PointClIP V2 Zhu et al. (2023). The $G(\cdot)$ pipeline has four steps.

**Voxelize.** First voxelized the point cloud space and projected the points belonging to the object into the voxels. Due to the occlusion properties of the point cloud, we simply assign the minimum depth value for the corresponding voxel.

**Densify.** Following PointCLIP V2, we densify the grid using a local minimum value pooling operation to improve visual continuity in sparse point clouds, ensuring that vacant voxels between sparse points are filled with reasonable depth values. This method, favoring minimum depth values, results in denser and smoother spatial representations, while keeping background voxels empty.

**Smooth.** To counteract potential artefacts from local pooling on 3D surfaces, we use a non-parametric Gaussian kernel for smoothing and noise filtering, achieving a balance between noise removal and edge sharpness preservation. In all experiments in our paper, we utilize the learnable smoothing as the few-shot classification in PointCLIP V2 with $1 \times 3 \times 3$ kernel size 3D convolution.

**Squeeze.** Finally, we compress depth to form a depth map to get a point cloud's depth map from a particular view.

## A.3 Details of Multi-view MAE

In our pipeline, we apply the basic 2D masked autoencoder (MAE) He et al. (2022) as a part of our pre-training backbone. Due to the limitation of the resource, we have to utilize the ViT-base Dosovitskiy et al. (2020) MAE. The MAE is pre-trained on the ImageNet1K dataset. The additional layers in the MAE encoder have the same architecture as the multi-head attention in the ViT-base.

## A.4 Details of the 2D-to-3D Mv-Swin

In Mv-Swin part, we utilize the multi-view attention and the Swin transformer blocks to extract the point cloud feature. We utilize very small dimensions 56 in swin blocks, and each block has $[2, 2, 6]$ and $[3, 4, 12]$ heads, respectively. The head dimension is set as 16, and the window size is set as 14. The point cloud features $f_{3D}$ in our experiments with $[bs, 1024]$ dimension.

## A.5 Details of 2D&3D Reconstruction Loss

We apply the MSE loss and Chamfer Distance loss to evaluate the reconstruction results of 2D and 3D respectively.

Mean Squared Error (MSE) loss, measures the average squared differences between the predicted images and the actual images. The mathematical representation of MSE loss is given by:

$$\text{MSE} = \frac{1}{N} \sum_{i=1}^{N} (y_i - \hat{y}_i)^2,$$

where $N$ is the number of samples, $y_i$ is the actual value of the $i$-th sample, $\hat{y}_i$ is the predicted value for the $i$-th sample. The goal of minimizing MSE loss is to find model parameters that reduce the average squared difference between predicted and actual iamges.

Chamfer Distance (CD) is a commonly used metric and loss function for comparing two point clouds. It measures the average closest point distance between two point sets, providing a symmetric and differentiable metric that quantifies how similar two point clouds are. The Chamfer Distance between two point clouds $A$ and $B$, where $A = \{a_1, a_2, ..., a_M\}$ and $B = \{b_1, b_2, ..., b_N\}$, can be defined as follows:

$$\text{CD}(A, B) = \frac{1}{|A|} \sum_{a \in A} \min_{b \in B} ||a - b||_2^2 + \frac{1}{|B|} \sum_{b \in B} \min_{a \in A} ||b - a||_2^2,$$

Table 6: The metrics between PointNet reconstruction and Mv-Swin 2D to 3D, where CD is scaled by $10^4$.

| Recon. Methods | CD ↓ |
|----------------|------|
| PointNet Recon. | 2.8 |
| Mv-Swin | 3.3 |

Table 7: Point cloud reconstruction metric with different blocks in the Mv-Swin, where CD is scaled by $10^4$.

| Mv-S Blocks | CD ↓ |
|-------------|------|
| Self-attention | 4.6 |
| w/o Mv-attention | 4.8 |
| w/ Mv-attention | 3.3 |

where $||a - b||_2$ denotes the Euclidean distance (L2 norm) between points $a$ and $b$, $|A|$ and $|B|$ denote the number of points in point clouds $A$ and $B$ respectively. In point cloud tasks, using Chamfer Distance as a loss function encourages the reconstructed point cloud to closely approximate the shape and distribution of the target point cloud.

### A.6 MORE ABLATION STUDY

**Mv-Swin Performance Evaluation.** In this section, we aim to validate the 3D representation capabilities of the 2D-to-3D reconstruction module, demonstrating that the reconstruction of 3D objects does not hinder the overall performance of the model. We utilize the Chamfer Distance (CD) for evaluating point cloud reconstruction as our metric and compare the multi-view reconstruction performance against the PointNet reconstruction model as a benchmark on the ShapeNet55 dataset. As shown in Tab. 6, our 2D-to-3D reconstruction module Mv-Swin gets 3.3 CD performance, which is close to the PointNet reconstruction performance. This indicates that the Mv-Swin 2D-to-3D reconstruction capability has achieved commendable performance, enabling the MV3D-MAE model to learn reasonable 3D representations.

**Mv-Attention in Mv-Swin.** Mv-Attention block is applied to aggregate multi-view depth image features in Mv-Swin to reconstruct the point cloud. In this ablation study Tab. 7, we remove the Mv-Attention blocks to evaluate the original Swin transformer capability to reconstruct the single point cloud object. Furthermore, when we substitute self-attention for Mv-attention, we observe that the performance is similar to that achieved when Mv-attention was removed. This demonstrates that the use of multi-view attention can assist Mv-Swin in better aligning 2D-3D features.

### A.7 FOR DOWNSTREAM TASKS

**Classification Task:** After obtaining the image features [B, 10, 197, 768], we extract the cls token values, resulting in [B, 10, 1, 768]. Since one 3D object corresponds to 10 depth images, we first compute the GroupAttention for each set of 10 cls tokens, then perform max pooling to obtain a [B, 1, 768] feature vector. This feature vector is then linearly transformed to produce a [B, 40] classification vector.

**Segmentation Task:** For the segmentation task, we take the dense part of the image features, which is [B, 10, 196, 768]. We then reverse map this feature to the 2D image and perform upsampling to get features of [B, 10, 224, 224, 768]. Using the back projection strategy from PointCLIP V2, we map these image features back to 2048 points using the saved coordinates of the point cloud on the image, resulting in [B, 10, 2048, 768] features. After max pooling, we obtain [B, 2048, 768] feature values. Finally, these features are linearly transformed to produce [B, 2048, 50] dense segmentation logits. This ensures that the dense features of the image are mapped onto the point cloud to facilitate segmentation.

