# OpenReview forum: "MV3D-MAE: 2D Pre-trained MAEs are Effective 3D Representation Learners"
_ICLR.cc/2025/Conference — Submitted to ICLR 2025_

### Official Review · Reviewer_Qz4N · 2024-10-17

**Soundness:** 3
**Presentation:** 2
**Contribution:** 3
**Rating:** 6
**Confidence:** 5

**Summary:**

This paper presents MV3D-MAE, a masked autoencoder framework that utilizes pre-trained 2D MAE models to enhance 3D representation learning. First, a single 3D point cloud is converted into a multi-view depth image. Based on the pre-trained 2D MAE model, the model is adapted to multi-view depth image reconstruction by integrating group attention and adding additional attention layers. A differentiable 3D reconstruction method, Mv-Swin, is then proposed, which learns 3D spatial representations by mapping the reconstruction results back to 3D objects without the use of camera poses. As a result, MV3D-MAE mitigates the differences between modalities by bi-directional conversion between 2D and 3D data and enhances the network's representational performance by utilizing a priori knowledge from pre-trained 2D MAEs.

**Strengths:**

1. The introduction section of the paper is better written and clearly motivated, and the key question of the paper is formulated by analyzing the existing research. ‘Can a method leveragepre-trained 2D MAE networks, facilitating mutual transformation between 2D and 3D, to achieve self-supervised pre-training for 3D representation?’
2. Unlike previous 2D-3D or 3D-2D mappings, the group attention module and MV-Swin module proposed in this paper can make the mapping process microscopic, which facilitates the information transfer between modalities.
3. From the description of the method section, this paper mainly adds network components to the existing 2D MAE model to make it suitable for 3D MAE learning. It is simpler in content, but has greater potential in application, which is in line with the idea of MAE.

**Weaknesses:**

1. It is recommended that the italicized mathematical symbols A and M in Section 3.2 be changed to uppercase Roman symbols. Moverover, it is suggested that the paper add formulas to show how 3D-2D and 2D-3D microprojections are realized.
2. Although the experimental part of the thesis is similar in kind to the related methods, it is suggested that comparison methods can be added. Also, it is suggested that the proposed method be validated on three variants of ScanObjectNN.
3. In the ablation experiments, $\lambda$ is the weighting term for the 3D reconstruction loss. In general, there is chance in such parametric experiments. It is proposed to further analyze how this hyperparameter affects the experiments. For example, the effect of 3D point cloud reconstruction when $\lambda$ is transformed.

**Questions:**

1. Why do you only do 2D masking for depth maps? According to all I have, point clouds can be projected as RGB images, and it is recommended to refer and cite CrossNet (TMM'2023) and Inter-MAE (TMM'2023), which fuel 3D self-supervised learning by extracting color image features. Also, it is optional to add them as comparison methods.
2. In Fig. 3, why should all the results obtained be the same when the query Q matrix is repeated and then maximized?

Overall, this is a practice-oriented work, which is not fully compatible with the ICLR theme. If the authors would consider my comments and make explanations and modifications, I might improve my score further. And vice versa.

---

> ### Author Response · Authors · 2024-11-23
> **Thanks for your review! Authors' feedback.**
>
> Thank you for your thoughtful review of our submission. Your feedback is invaluable, and we are grateful for your constructive comments, here is the answer to your questions:
>
> **For Comment 1: Italicized mathematical symbols and more formulas for 2D-3D and 3D-2D.**
>
> Thanks for your suggestions! We will soon update the notation to use uppercase Roman symbols in the paper. Additionally, we will clarify the projection process more explicitly using equations.
>
> **For Comment 2: More experiment results.**
>
> In line with the previous reviewers, we have included comparisons with ACT, Point-JEPA, and PointGPT. We also extended our experiments by testing three ScanObjectNN experiments based on the original setup.
>
> **Table 6. More performance comparison of all additional.**
> | **Method**   | **ModelNet40 Accuracy (%)** | **ScanObjectNN Accuracy (%)** |
> |-|-|-|
> | I2P-MAE  | 93.7                        | 91.57                        |
> | ACT     | 93.50                       | 89.16                         |
> |  Point-JEPA  | 93.8                     | 86.6                         |
> |Point-JEPA  | 94.0                    | 90.0                     |
> | **Ours**     | 94.1                        |     89.85                 |
>
>
> We also tested three ScanObjectNN experiments based on the existing framework.
> The results on ScanObjectNN OBJ-B, OBJ-ONLY, and PB-T50-RS are as follows:
>
> **Table7. More performance comparison on ScanObjectNN.**
> | Method       | OBJ-B  | OBJ-ONLY | PB-T50-RS |
> |-|-|-|-|
> | ACT          | 91.22  | 89.16    | 85.81     |
> | Point-MAE    | 90.02  | 88.29    | 85.18     |
> | Joint-MAE    | 90.94  | 88.86    | 86.07     |
> | Point-M2AE   | 91.22  | 88.81    | 86.43     |
> | I2P-MAE      | 94.15  | 91.57    | 90.11     |
> | **Our Method**| 91.63  | 89.85    | 86.57    |
>
> As shown in the table, I2P-MAE achieves the best performance across all tasks. However, our method consistently delivers competitive results, surpassing all other methods, including ACT, Point-MAE, Joint-MAE, and Point-M2AE. The performance of our model is very close to I2P-MAE, and while we do not outperform I2P-MAE, our method still achieves remarkable results.
>
> **For Comment 3: How λ influences the performance.**
>
> To test how λ influences the performance of the pre-training. We do more experiments in training. We use the CD (Chamfer Distance) metric to compare the 3D reconstruction results. The results are shown in the table below:
>
> **Table 8.  Performance on Reconstruction of Point Cloud with Different lambda.**
>  |λ|CD Metric | Cls on ModelNet40
> |-|-|-|
> |0|-|91.4|
> |25|3.3|94.1|
> |50 |3.9|93.1|
> |100 |4.2|92.2|
>
> When we set the value of λ to 0, 3D reconstruction does not occur. However, when we set λ to 25, the best metrics are achieved for reconstruction. As we increase the weight of λ, the performance of the 2D reconstruction model deteriorates, which in turn leads to a decline in 3D reconstruction results.
> And the Table 8 proves that the performance of 3D reconstruction impacts the MAE component's effectiveness in downstream 3D tasks. This further highlights the importance of the learning process of the 3D reconstruction module in the MV3D-MAE model.
>
> **For Question 1: Can we introduce RGB image to the pipeline?**
>
> We really appreciate your question, as it aligns with one of our main concerns. After completing this work, we conducted further discussions regarding this issue.
> If you wish to adopt the same paradigm as our network—first projecting 3D into 2D, performing MAE-related operations on the 2D images, and finally fusing the 2D images back into a 3D model—this approach is not limited to 3D point clouds and could be applied to other formats.
> During this work, we were unable to find a suitable method to effectively merge multiple RGB images into a cohesive 3D object. However, we have recently come across methods, such as *3D Reconstruction with Spatial Memory*, that could potentially achieve this.
> We are currently conducting related research in this area and hope to achieve even better results in the future.
>
> **For Question 2: how the maxpooling work in the Mv-Swin?**
>
> Thank you for this question! We adopted the max-pooling strategy to extract the most salient feature for each object. In our approach, the input consists of multi-view images with 10 viewpoints per object. Therefore, the number of samples in each batch is always a multiple of 10. For instance, 40 depth images represent 10 views of 4 distinct objects. When extracting features for each object, we first compute the most salient feature across the 10 views and replicate it 10 times to maintain the batch size. This replicated feature is then used as the query. While the replication may appear as repeating the same feature 10 times (with no change in color), the final feature is obtained by applying max-pooling across the views. Not sure if we misunderstood your point, but we believe there are no errors in the colors in Figure 3.

---

> > ### Comment · Reviewer_Qz4N · 2024-11-26
> > **Reviewer's Response**
> >
> > Thanks to the authors for their responses, which addressed most of my concerns. As a result, I am willing to raise my rating to 6. The reason is: considering that the work may be concerned by other reviewers and the originality is not a perfect match with the theme of ICLR, I cannot fully recommend it.
> >
> > I hope you can revise the content according to the reviewers' comments and add the missing references. Good luck!

---

> > > ### Author Response · Authors · 2024-12-02
> > > **Author response.**
> > >
> > > Thank you for your recognition and score improvement!
> > >
> > > Although our approach does not rely heavily on mathematical formulations, our training paradigm is distinct from previous 2D-3D pretraining models. This paradigm provides a more general solution for scaling the model. We will continue to explore this promising direction further.

---

### Official Review · Reviewer_B9Fi · 2024-10-27

**Soundness:** 3
**Presentation:** 3
**Contribution:** 2
**Rating:** 5
**Confidence:** 3

**Summary:**

The paper introduces MV3D-MAE, a framework designed to improve 3D representation learning by leveraging pre-trained 2D models. Since acquiring 3D data is more expensive and time-consuming than 2D data, the authors propose converting 3D point clouds into multi-view depth images. They enhance a pre-trained 2D MAE model with group attention and additional layers for reconstructing these depth images. Their method, Mv-Swin, then maps these reconstructions back into 3D without using camera poses, facilitating better 3D spatial learning. This approach significantly boosts performance in few-shot classification, achieving state-of-the-art results in various tasks such as classification and segmentation across synthetic and real-world datasets.

**Strengths:**

1. The paper is clearly written.
2. The proposed method is effective, which leverages a pretrained 2D model for 3D learning and introduces MV-Swin for smooth 2D-to-3D transitions, resulting in superior few-shot classification performance.

**Weaknesses:**

1. The experiments are conducted mainly on ModelNet and ScanObjectNN, which are quite simple. How about the results on ShapeNet55-34 and ShapeNetPart? It would be interesting to see whether the proposed method is effective for the long-tail classification problem.
2. Missing comparison to the latest SOTA:
- Point-JEPA: A Joint Embedding Predictive Architecture for Self-Supervised Learning on Point Cloud. [arXiv 2404.16432]
- PointGPT: Auto-regressively Generative Pre-training from Point Clouds. [NeurIPS 2023]

**Questions:**

Please refer to the weaknesses.

---

> ### Author Response · Authors · 2024-11-23
> **Thanks for your review! Authors' feedback.**
>
> Thank you for your thoughtful review of our submission. We appreciate your recognition of our pre-training method for learning effective representations from point clouds and the validation of our approach using 2D pre-trained models. Your feedback is invaluable, and we are grateful for your constructive comments, here is the answer to your questions:
>
> **For comment1: Why we not fine-tuning on ShapeNet55-34 and long-tail data.**
>
> Our pretraining experiments were conducted on ShapeNet55-34, so we may not conduct further fine-tuning on this dataset. However, our segmentation task is carried out on the ShapeNetPart dataset, as shown in Table 3.
>
> There is limited research on long-tail data in point cloud datasets. After analyzing the existing data, we found it difficult to construct a long-tail dataset that meets the conditions for training or fine-tuning. We appreciate the reviewer's suggestion and will consider further research on this topic in the future.
>
> **For comment2: More comparison with Point-JEPA and PointGPT-S.**
>
> Compared with Point-JEPA:
>
> We compared the performance of our model with Point-JEPA and PointGPT, as shown below:
>
> - **On the ModelNet40 classification task**:
>   - Our model achieved **94.1** accuracy without voting, compared to Point-JEPA's **93.8** (without voting).
> - **On the ScanObjectNN classification task**:
>   - Point-JEPA achieved approximately **86.6 ± 0.3** accuracy (as reported on its GitHub page), which is lower than our model's performance of **89.85**.
> - **On the segmentation task**:
>   - Our segmentation performance reached **85.5**, slightly lower than Point-JEPA's **85.8 ± 0.1**.
>   - This slight decrease is expected, as our method requires projecting 2D multi-view features back into 3D space in the latent space for segmentation, introducing additional complexity.
>
> Compared with PointGPT-S:
>
> We compared our model with PointGPT-S, which has the closest parameter count to our model:
> - **On ModelNet40**:
>   - PointGPT-S achieved **94.0** accuracy, close to our model's performance.
> - **On ScanObjectNN**:
>   - PointGPT-S achieved **90.0** accuracy, comparable to our result.
> - **On the segmentation task**:
>   - PointGPT-S only achieved **84.1**, which is significantly lower than our model's **85.5**, demonstrating our superior performance in this task.
>
> **Table5. More performance comparison.**
> | Task                        | Our Model  | Point-JEPA | PointGPT-S |
> |-----------------------------|------------|------------|------------|
> | **ModelNet40 Classification**| 94.1       | 93.8       | 94.0       |
> | **ScanObjectNN Classification** | 89.85      | 86.6 | 90.0       |
> | **Segmentation on OBJ-ONLY** | 85.5       | 85.8 | 84.1       |

---

> > ### Comment · Reviewer_B9Fi · 2024-11-29
> >
> > Thanks for your rebuttal.
> >
> > The authors have addressed most of my concerns.
> > 1. Thank you for including the comparison to Point-JEPA and PointGPT-S. Please make sure to add these results to the revised PDF.
> > 2. I am still unclear on why introducing additional complexity leads to a decrease in segmentation performance.

---

> > > ### Author Response · Authors · 2024-12-02
> > > **Author response.**
> > >
> > > Thanks for your comments!
> > >
> > >  We will update the results in the revised paper.
> > >
> > > Regarding your concern about why segmentation performance is not good enough:
> > >
> > > Consistent with the PointCLIP method we referenced, we stored the positional mapping information from the 3D projection to 2D to help maintain the correspondence between each point and its 3D location during segmentation. When performing 3D segmentation, we reverse-map the complete 2D features back into the 3D space using the stored positional information. However, this mapping process inherently introduces some loss, and the reverse-mapping of features after the encoder lacks an effective reconstruction learning process. Instead, segmentation training is directly conducted on the 2D features. Following the standard approach, we utilized only the encoder portion of the 2D-MAE for this process and did not perform fine-tuning on the 3D features for segmentation training.

---

> > > > ### Comment · Reviewer_B9Fi · 2024-12-02
> > > >
> > > > I see. Thanks for your explanation.
> > > > It seems that the PDF has not been updated.

---

> > > > > ### Author Response · Authors · 2024-12-03
> > > > > **Author's response**
> > > > >
> > > > > Thanks for your reply! Now, the PDF version cannot be updated, so we will send the revised paper to the conference if it is accepted.

---

### Official Review · Reviewer_hF11 · 2024-11-01

**Soundness:** 2
**Presentation:** 2
**Contribution:** 2
**Rating:** 5
**Confidence:** 5

**Summary:**

This paper leverages a pre-trained 2D MAE in point cloud representation learning by for- and back-ward rendering of multi-view depth images. The proposed method achieves SoTA performances across several tasks. Some module analysis and key results are missing.

**Strengths:**

1.  Testing on incomplete data and cross-modality retrival is a nice trial.
2.  By using a pre-trained 2D MAE, the model significantly enhances 3D feature learning capacity, yielding SoTA performances.

**Weaknesses:**

1. The module to reconstruct multi-view depth images to point cloud is waird and somewhat redundant, as a simple depth projection might yield similar results; the authors should validate this experimentally by replace it with mv-depth image projection. The claim of no pose information requirement (L86) is also questionable, as poses are manually set during depth map generation.
2. Training costs of the proposed method is high, which may harm it generalizability. Please provide and compare the model parameter amount and FLOPs.
3. The simulation of real-word data is still not enough. Trend analysis between the noise-level or completeness and the final performances are needed.
4. Qualitative and quantitative results are desired, espically the failure case analysis.

**Questions:**

Please refer to the weakness.

---

> ### Author Response · Authors · 2024-11-23
> **Thanks for your review! Authors' feedback [1/2].**
>
> Thank you for your thoughtful review of our submission. Your feedback is invaluable, and we are grateful for your constructive comments, here is the answer to your questions:
>
> **For Comment1: Why do we need to reconstruct the point cloud and the revision of “no pose”.**
>
> Thank you for your comment. The design of a novel Mv2D depth image-to-3D point cloud structure was carefully considered. The mapping process of converting a 3D point cloud object into a smooth point cloud structure is irreversible, as it involves not only projection but also steps like densification and smoothing (see Appendix A.2). Using a simple multi-view 2D-to-3D projection and merging process would make it difficult for depth images reconstructed through MAE to form a coherent 3D object. This would undermine the supervision of the 3D loss and impair the model’s capability to learn the relationship between 2D and 3D structures. Hence, our 2D-to-3D projection module is essential.
>
> Regarding "no pose information requirement," we used fixed parameters for stable 3D synthesis and applied random rotations during pretraining to improve robustness. We acknowledge your critique and will revise our wording to clarify: "Our model can perform 2D-to-3D mapping for depth images captured under fixed relative positions, without requiring explicit pose conditioning." This will be updated in the paper.
>
> **For Comment2: Parameter amount and FLOPs.**
>
> In the pre-training stage, we mainly train the parameters in GroupAttn and the Add layers in MAE, with a total parameter count of 28.35 million. However, when using the MV-Swin encoder-decoder for 3D reconstruction, the model's parameter count reaches 46.25 million, which means that the 2D-3D model have 17.9 million parameters.
>
> Point-MAE and Point-BERT have a parameter count of 22.1 million. These models do not involve 2D-3D conversion and mapping processes, resulting in a smaller parameter size.
>
> For downstream tasks, we do not need to use the MV-Swin structure for 3D reconstruction, so the speed is relatively fast. However, the segmentation head requires mapping dense features to point clouds. Although this part has only 11 million parameters, the parameter-free 2D image to 3D point cloud mapping process still takes a lot of time, causing the segmentation head to still require 90.8ms.
> The training time cost for each part in different stages can be referred in Table 3:
>
> Table 3. Training time cost for each part.
> | Stage           | Step                        | Time (ms) |
> |-|-|-|
> | Pre-training     | Point cloud to depth maps  | 1.5       |
> |                  | MAE encoder-decoder        | 13.8      |
> |                  | Depth maps to point cloud  | 8.7    |
> | Classification   | Point cloud to depth maps  | 1.4       |
> |                  | MAE encoder                | 8.3       |
> |                  | Classifier head            | 1.0       |
> | Segmentation     | Point cloud to depth maps  | 1.4       |
> |                  | MAE encoder                | 8.5       |
> |                  | Segment head               | 90.8      |
>
> As the comparison, the Point-MAE and Point-BERT training pipeline cost ~94ms and ~106ms. The classification head cost ~1ms, segmentation head cost ~86ms. Our model’s MAE module has an efficient throughput time.
>
> **For comment3: The analysis of partial point cloud experiments.**
>
> Based on Table 2 in the review, we further analyze why our performance surpasses that of Point-MAE and I2P-MAE on incomplete datasets.
>
> First, we examine the data patterns in the PCN dataset. Since this dataset originates from the real world and is obtained using depth scanners, the missing patterns in the data are often distinguishable to the human eye.
>
> As shown in the Figure1 in the [link](https://anonymous.4open.science/r/MV3D-MAE-2D-Pre-trained-MAEs-are-Effective-3D-Representation-Learners-Rebuttal-6445/Figure1%20Differences%20between%20Real%20world%20PCD%20and%20Cropped%20PCD.png), when our model has access to three or more visible viewpoints, its performance remains consistently reliable. This demonstrates that, after fine-tuning, our model can classify objects based on certain visible viewpoints of an incomplete object.
>
> For the ModelNet40-Crop dataset, which contains incomplete data, some viewpoints remain visible. After fine-tuning, our model achieves good performance. However, without fine-tuning, none of the three methods can deliver usable performance.
>
> Therefore, we believe that all existing methods struggle to directly generalize to incomplete and noisy point cloud datasets. To better adapt to incomplete point clouds in real-world scenarios, in addition to direct data collection, we are particularly interested in learning a pattern of incomplete point clouds from real-world data. This pattern could be used to degrade complete point clouds into incomplete ones, thereby aiding point cloud classification algorithms in better generalizing to real-world environments.

---

> ### Author Response · Authors · 2024-11-23
> **Thanks for your review! Authors' feedback [2/2].**
>
> **For comment4: Failure analysis**
>
> Analysis of Classification Errors for the Top 5 Categories
> Table4. Misclassification Analysis Table
> | Category         | Total Samples | Misclassified Samples | Error Rate | Misclassification Details                                                       |
> |-|-|-|-|-|
> | **Flower Pot**   | 20            | 12                     | 60.00%     | Plant: 6 times (50.00%), Vase: 5 times (41.67%), Cup: 1 time (8.33%)           |
> | **Night Stand**  | 86            | 42                     | 48.84%     | Dresser: 26 times (61.90%), Table: 7 times (16.67%), TV Stand: 4 times (9.52%) |
> | **Cup**          | 20            | 6                      | 30.00%     | Vase: 5 times (83.33%), Bowl: 1 time (16.67%)                                  |
> | **Stool**        | 20            | 5                      | 25.00%     | Chair: 3 times (60.00%), Vase: 1 time (20.00%), Table: 1 time (20.00%)         |
> | **Table**        | 100           | 25                     | 25.00%     | Desk: 22 times (88.00%), Bench: 2 times (8.00%), Night Stand: 1 time (4.00%)   |
>
> In our analysis of the 2,468 test objects, with an overall classification accuracy of 94.1%, a total of 145 objects were misclassified. Among these errors, the Flower Pot category consistently exhibited the highest misclassification rate. And also the model is hard to classify the Night Stand and dressers. So we illustrate the figures of Flower Pot and Plant, the Night Stand and Dresser in the [Figure2 link](https://anonymous.4open.science/r/MV3D-MAE-2D-Pre-trained-MAEs-are-Effective-3D-Representation-Learners-Rebuttal-6445/Figure2%20Classes%20Visualization.png)
>
> Error Analysis and Observations
>
> The Flower Pot category frequently suffers from misclassification, often being confused with Plant, Vase, or Cup. This misclassification stems from the inherent similarity between Flower Pot and Plant, as both typically consist of two main parts: a vase or pot and a plant. Figure 2 in the referenced link highlights this issue, showing that many objects in the Flower Pot category share indistinguishable features with objects in the Plant category, making it challenging for the model to classify accurately. Moreover, ambiguities in the dataset's labeling process further exacerbate this issue, as the distinction between Flower Pot and Plant categories is poorly defined.
>
> A similar challenge arises between the Night Stand and Dresser categories. These two categories share significant visual overlap, and size often serves as a critical factor in human judgment for distinguishing them—dressers are generally larger than nightstands. However, the preprocessing step in the dataset normalizes object sizes by resizing them to a uniform scale, which removes this crucial contextual information. Consequently, the model struggles to differentiate between these two categories due to the loss of this distinguishing factor.
>
> Conclusions and Thinking Solutions
>
> From the above analysis, we conclude that existing methods face significant challenges when classifying objects in point cloud datasets without color information. Categories like Flower Pot vs. Plant, Night Stand vs. Dresser, and Chair vs. Stool are highly similar, and in some cases, objects are virtually identical but classified into different categories. This ambiguity in the dataset undermines classification accuracy. Furthermore, the dataset's resizing process eliminates size-related distinctions, which are particularly critical for differentiating categories like Dresser and Night Stand.
> To address these challenges, we suggest two potential solutions. First, incorporating color information into point clouds could significantly improve classification performance, as color often provides critical cues for distinguishing between similar categories. For instance, green hues could help identify plants, while unique patterns or colors could differentiate flower pots from vases. Second, leveraging Vision-Language Models (VLMs) could further enhance classification accuracy by integrating visual features with semantic textual descriptions of objects. By combining color information with VLMs, models can better generalize to ambiguous cases and improve classification in real-world settings.

---

> > ### Comment · Reviewer_hF11 · 2024-12-01
> > **Reviewer's Feedback**
> >
> > Thank you for the detailed responses to my initial review. While some aspects of the rebuttal provide additional clarity, the key concerns remain inadequately addressed, particularly regarding efficiency, generalizability, and comprehensive comparisons. As such, my overall conclusion remains unchanged.
> >
> > 1. Module Necessity and Simplifications:
> > The rebuttal explains the design of the depth-to-point cloud reconstruction module but does not include an ablation study to validate its indispensability. Simpler alternatives like depth projection should be tested to demonstrate that the complexity is warranted.
> > 2. Insufficient Dataset Coverage:
> > The experiments remain limited to relatively simple datasets (ModelNet and ScanObjectNN), and the proposed method has not been validated on more challenging datasets like ShapeNet55-34. This limits confidence in the method's robustness for real-world scenarios or long-tail classification problems.
> > 3. Missing Comparisons:
> > Comparisons with ACT, PointGPT-S, and Point-JEPA are acknowledged, but important baselines such as ReCon (Contrast with Reconstruct: Contrastive 3D Representation Learning Guided by Generative Pretraining)  remain missing. Without a comprehensive set of comparisons, it is difficult to evaluate the true contribution of the work.
> > 4. Segmentation Performance Drop:
> > The rebuttal briefly attributes the segmentation performance drop to added complexity but does not provide sufficient analysis or evidence to support this claim. Further investigation into this issue is necessary.

---

> > > ### Author Response · Authors · 2024-12-02
> > > **Author response.**
> > >
> > > **For comment1:Module Necessity and Simplifications**
> > >
> > > To intuitively understand why direct 3D projection cannot be applied to reconstructed depth maps, we quantitatively demonstrate the results of direct projection. Specifically, we reconstruct multi-view depth maps as point clouds through direct projection, then merge the point clouds and compute the CD (Chamfer Distance) loss against the ground truth. Compared to the CD metric of 3.3 achieved using the Mv-Swin method, the CD metric for direct projection is 24.7, rendering it practically unusable.
> > > This poor performance arises because the reconstructed depth maps are not fully normalized, and the numerical precision of the reconstructed depth differs significantly from that of traditional depth images. Consequently, direct projection introduces significant errors. Moreover, since this process is not learnable, direct projection fails to provide supervision for the 3D component of the model. As a result, the model cannot effectively learn 3D representations or assist in the learning of 2D features. This is evident in our ablation study, where setting the loss coefficient λ\lambda to 0 leads to degraded model performance.
> > > From these findings, we can conclude how 2D-to-3D projection can aid 2D-MAE in learning the overall shape of objects:
> > > Ensuring learnability and trainability of the 3D model is essential. By incorporating the reconstruction of 3D models into the training process, the model can learn effectively, which is impossible with direct depth projection.
> > > The reconstructed depth maps and directly projectable depth maps differ in data type and features from the original depth maps. Direct projection fails to recover the original 3D objects. In contrast, a learnable process enables the model to learn the missing data during projection, resulting in better reconstruction outcomes (as shown in Figure 6 of the paper).
> > >
> > >
> > >
> > > **For Comment2: Insufficient Dataset Coverage**
> > >
> > > Our pretraining experiments were conducted on ShapeNet55-34, so we avoid further fine-tuning on this dataset. Similar to other related works such as Point-MAE, ACT, and PointGPT, our model is tested on standard benchmark datasets. Additionally, we have extended our evaluations to a broader range of datasets, including experiments on incomplete real-world data and synthetic data, demonstrating the versatility and robustness of our approach.
> > >
> > >
> > > **For Comment3: Missing Comparisons**
> > >
> > > The latest results have been included in Tables 9 and 10 for your reference.
> > >
> > >
> > > **For Comment4:Segmentation Performance Drop**
> > >
> > > Consistent with the PointCLIP method we referenced, we stored the positional mapping information from the 3D projection to 2D to help maintain the correspondence between each point and its 3D location during segmentation. When performing 3D segmentation, we reverse-map the complete 2D features back into the 3D space using the stored positional information. However, this mapping process inherently introduces some loss, and the reverse-mapping of features after the encoder lacks an effective reconstruction learning process. Instead, segmentation training is directly conducted on the 2D features. Following the standard approach, we utilized only the encoder portion of the 2D-MAE for this process and did not perform fine-tuning on the 3D features for segmentation training.

---

### Official Review · Reviewer_ev39 · 2024-11-04

**Soundness:** 2
**Presentation:** 2
**Contribution:** 2
**Rating:** 5
**Confidence:** 4

**Summary:**

This work introduces MV3D-MAE, a novel masked autoencoder framework aimed at enhancing 3D representation learning by leveraging pre-trained 2D models. The authors address the challenge of costly 3D data acquisition by converting 3D point clouds into multi-view depth images, enabling the use of a pre-trained 2D MAE model. MV3D-MAE includes a differentiable 3D reconstruction module that reconstructs the depth images back into 3D point clouds without camera poses, thus enabling bidirectional transformation between 2D and 3D modalities.

**Strengths:**

1, The writing is clear, and easy to follow.
2. The experiments conducted are thorough and comprehensive.

**Weaknesses:**

1, The framework does not seem to make sense to me. In my view, it resembles a 2D encoder pre-training framework rather than a 3D encoder pre-training one, as its ultimate aim is to use multi-view 2D images to reconstruct 3D objects.

2, The results are incremental. Despite leveraging complex frameworks, the fine-tuning results on ScanObjectNN fall short compared to I2P-MAE, another 3D self-supervised learning method designed to utilize 2D foundation models. Additionally, ACT [1], a highly related approach, is not compared in this paper.

3, The comparison is unfair. The architecture of this method differs from the Point-MAE baseline. A fair comparison should involve Point-MAE with the same network architecture.

[1] Autoencoders as Cross-Modal Teachers: Can Pretrained 2D Image Transformers Help 3D Representation Learning?

**Questions:**

How is the model fine-tuned? Which components are retained, and which are discarded during the fine-tuning process?

---

> ### Author Response · Authors · 2024-11-23
> **Thanks for your review! Authors' feedback.**
>
> Thank you for your detailed review and valuable feedback on our paper. We highly value your constructive comments:
>
> **For comment1: Why we use 2D pre-trained MAE for 3D tasks?**
>
> Our goal is to leverage 2D pre-trained models to enhance the performance of 3D tasks. Acquiring 3D objects directly is often complex and challenging, whereas obtaining uniformly formatted 2D images is significantly more convenient. Therefore, exploring how 2D models can assist 3D models in completing tasks more effectively is a highly valuable research direction.
>
> In our experiments, the pre-trained 2D MAE has already learned extensive prior knowledge about images, including object edges and texture information. The features learned by the 2D MAE also help our model capture the textures and object features in depth images.
>
> As shown in Table 5 of the ablation study, there is a significant performance gap between the model without the pre-trained 2D MAE and the one with it. This highlights the effectiveness and necessity of using 2D pre-trained models in 3D tasks. Moreover, the use of pre-trained 2D MAE is scalable. Although we achieved the current performance using only a ViT-base model, due to resource limitations, we have not yet tested larger MAE models. We plan to further extend our work in this direction.
>
> **For comment2，Compare with I2P-MAE and ACT.**
>
> Our method indeed has some performance gaps compared to I2P-MAE on ScanObjectNN, but it achieves superior performance on ModelNet40 with an accuracy of 94.1 compared to I2P-MAE's 93.7. Moreover, our proposed method introduces a distinct 2D-3D training paradigm different from I2P-MAE, allowing for better utilization of 2D MAE capabilities and offering stronger scalability.
> Regarding the ACT method, our classification model outperforms ACT, which achieves 93.50 on ModelNet40. Furthermore, compared to ACT's 89.16 performance on ScanObjectNN without data augmentation, our model demonstrates an advantage with an accuracy of 89.85.
>
> **Table 1.  More comparison on downstream tasks.**
> | **Method**   | **ModelNet40 Accuracy (%)** | **ScanObjectNN Accuracy (%)** |
> |-|-|-|
> | **Ours**     | 94.1                        |     89.85                 |
> | **I2P-MAE**  | 93.7                        | 91.57                        |
> | **ACT**      | 93.50                       | 89.16                         |
>
> **For comment3，Further comparison with I2P-MAE on partial point cloud dataset.**
>
> In addition to using Point-MAE, we have also included I2P-MAE for further comparison on the partial point cloud dataset. The performance comparison is as follows:
>
> **Table2.  More comparison of performance on Incomplete Datasets.**
> |Dataset|Our Model(%)|Point-MAE (%)|I2P-MAE (%)|
> |-|-|-|-|
> |PCN dataset|92.2%| 91.1%|92.2%|
> |ModelNet40-Crop (fine-tuned)|92.04%| 90.41%|91.74%|
> |ModelNet40-Crop (Generalized)|67.3%|69.4%|68.8%|
>
> This demonstrates that our model achieves performance comparable to I2P-MAE on the PCN dataset. Furthermore, after fine-tuning on the ModelNet40-Crop dataset, our model outperforms I2P-MAE. However, when using a generalized model, the performance of our model drops to an unusable state, with accuracy falling below 70%.
>
> **For Question: How we fine-tuned and what components are retained?**
>
> Thanks for your questions！
>
> The specific fine-tuning architectures for classification and segmentation are detailed in the final section of the appendix. We further explain how we perform fine-tuning during the process as follows:
>
> During fine-tuning, we retain the 3D-to-2D projection module and the 2D encoder module of MV3D-MAE, while the 2D decoder module and the 2D-to-3D reconstruction module are discarded. All parameters are trainable during the classification and segmentation processes. The descriptions of the classification and segmentation heads have shown in the Appendix A.7.

---

> > ### Comment · Reviewer_ev39 · 2024-11-26
> >
> > Thank you to the author for the detailed response. However, my primary concerns remain unaddressed, particularly regarding the comparison and methodology. While it is well-known that leveraging foundation models can enhance performance, this paper does not adequately address critical aspects.
> >
> > For instance, an important paper, ReCon [1], which is highly relevant, is not included in the comparisons. Additionally, the classification results on ModelNet lack clarity, as results with and without voting are not separately reported. Similarly, on the ScanObjectNN dataset, results for the OBJ-BG, OBJ-ONLY, and PB-T50-RS splits are incomplete—only the OBJ-ONLY split is provided.
> >
> > The authors claim that the motivation for this work is to utilize 2D MAE pre-trained models. However, the benefit of using the 2D MAE model compared to existing methods (e.g., I2P-MAE and ReCon) is not clearly demonstrated in this paper. I do not see any specific reason 2D MAE mode is needed or can provide better representations. Furthermore, this work employs the ViT-B model, which introduces another issue of unfair comparison. Most existing works use ViT-S models by default. When ViT-B models are used, it is standard practice to explicitly note this, as ViT-B models generally achieve improved performance over ViT-S models.
> >
> > I hope these concerns can be addressed to strengthen the paper.
> >
> >
> > [1] Contrast with Reconstruct: Contrastive 3D Representation Learning Guided by Generative Pretraining

---

> > > ### Author Response · Authors · 2024-12-02
> > > **Author respones**
> > >
> > > Thanks for your further comments!
> > >
> > > We add experiments comparing our method with Recon in Table 9, and since the voting technique is specific to point cloud classification, and our method works on depth images, the comparison results w/o voting.
> > > In Table 9,  It can be observed that Recon performs exceptionally well on ScanObjectNN, surpassing typical methods. Based on conclusions drawn from "Thanks for your review! Authors' feedback [2/2]. Conclusions and Thinking Solutions," using language models and precise text annotations can significantly improve the classification of ambiguous 3D object categories. Recon employs such methods with additional text features, which considerably enhance model performance. However, this makes a direct comparison with our method unfair. We will explore incorporating text annotations in the future to improve text-2D-3D alignment and enhance the performance of 3D models. Additionally, our model has only 22.8M parameters for classification, significantly smaller than Recon’s 43.6M parameters. It is also important to note that both I2P-MAE and Recon default to using ViT-Base models, not ViT-Small, which further distinguishes their setups from ours.
> > >
> > >
> > > Compared to Recon, which incorporates image and text data for contrastive learning, and I2P-MAE, which uses image data as a mask guide to extract more effective 3D features, our approach of leveraging 2D pre-trained models is more intuitive and easier to scale for larger deployments. Specifically, our method directly explores whether 2D pre-trained models can effectively handle 3D tasks, aligning perfectly with our title: MV3D-MAE: 2D Pre-trained MAEs are Effective 3D Representation Learners.
> > > Unlike I2P-MAE and Recon, our approach fundamentally differs in its goal. We aim to directly utilize the 2D-MAE structure for 3D tasks, rather than treating 2D features as auxiliary inputs to enhance the performance of 3D models. Through the combined processes of 3D-to-2D and 2D-to-3D training, we demonstrate that 2D-MAEs can effectively learn 3D object features.
> > > From our experiments, it is evident that when the 2D-MAE pre-trained weights are not used, the model achieves an accuracy of 90.8. In contrast, when using the pre-trained weights, the accuracy improves significantly to 93.0. This substantial gap clearly demonstrates that 2D pre-trained models can effectively facilitate the learning of 3D representations.
> > >
> > >
> > >
> > > **Table 9. More performance comparison of all additional.**
> > > | **Method**   | **ModelNet40 Accuracy (%)** | **ScanObjectNN Accuracy (%)** |
> > > |-|-|-|
> > > | I2P-MAE  | 93.7                    	| 91.57                    	|
> > > | ACT 	| 93.50                   	| 89.16                     	|
> > > |  Point-JEPA  | 93.8                 	| 86.6                     	|
> > > |Point-JEPA  | 94.0                	| 90.0                 	|
> > > |Recon|94.1| 93.63 |
> > > | **Ours** 	| 94.1                    	| 	89.85             	|
> > >
> > >
> > > **Table10. More performance comparison on ScanObjectNN.**
> > > | Method   	| OBJ-B  | OBJ-ONLY | PB-T50-RS |
> > > |-|-|-|-|
> > > | ACT      	| 91.22  | 89.16	| 85.81 	|
> > > | Point-MAE	| 90.02  | 88.29	| 85.18 	|
> > > | Joint-MAE	| 90.94  | 88.86	| 86.07 	|
> > > | Point-M2AE   | 91.22  | 88.81	| 86.43 	|
> > > | I2P-MAE  	| 94.15  | 91.57	| 90.11 	|
> > > | Recon 	| 95.18  | 93.63           |90.63 	            |
> > > | **Our Method**| 91.63  | 89.85	| 86.57	|

---

> > > > ### Comment · Reviewer_ev39 · 2024-12-03
> > > >
> > > > Thank you to the authors for their detailed response. In the ReCon paper, the ablation study focusing solely on the image modality reports a result of 90.18 on the PB-T50-RS split (compared to 86.57 in this work). Additionally, the results reported in ACT—93.29, 91.91, and 88.21—are also notably higher than those achieved in this study. This highlights a significant performance gap between this work and other 3D models trained using foundation models.
> > > >
> > > > While the authors argue that utilizing the 2D-MAE structure for 3D tasks is more intuitive and scalable for larger deployments, no supporting evidence is provided to substantiate this claim. It raises the question of why MAE with ViT-L or ViT-G was not directly applied to the Objaverse dataset for pre-training.
> > > >
> > > > Given the considerable performance gap between this work and other models trained on foundation models, I will maintain my original score.

---

> > > > > ### Author Response · Authors · 2024-12-03
> > > > > **Author's response**
> > > > >
> > > > > Thanks for your comments. In our response, we discussed that Recon gets a good performance because of the text-2D-3D dataset. Although our model has a performance gap between the ACT and Recon, the proposed pipeline still has the novelty and is worthy to explore. We will add the RGB image data and scale the ViT model for further training in the future.

---

### Comment · Area_Chair_qf6A · 2024-11-27
**Reminder: Last day for author feedback**

This is a reminder that today is the last day allotted for author feedback. If there are any more last minute comments, please send them by today.

---

### Meta-Review · Area_Chair_qf6A · 2024-12-19

**Metareview:**

The authors propose a Masked Autoencoder (MAE)-based 3D representation learning method. The main contribution is in leveraging pre-trained 2D models by converting 3D point clouds into depth maps.

We have read the referee reports and the author responses. Main concerns include the ablation study of the proposed depth-to-point cloud reconstruction module, additional experiments on more difficult datasets, lack of comparisons with state of the art methods, and lack of evidence for the claim that 2D pretraining of is more intuitive and scalable. The authors have provided extensive results, but were unable to fully address the comments of ev39, hF11, and Qz4N. While the authors seem to have addressed comments from B9Fi, after discussion, B9Fi ultimately lowered their score. A consensus was reached that the manuscript is not suitable for acceptance at the current state. While we acknowledge that the authors have provided comprehensive experiments, given the recommendation, we encourage the authors to incorporate the changes to the manuscript.

**Additional Comments On Reviewer Discussion:**

Main concerns include the ablation study of the proposed depth-to-point cloud reconstruction module, additional experiments on more difficult datasets, lack of comparisons with state of the art methods, and lack of evidence for the claim that 2D pretraining of is more intuitive and scalable. The authors have provided extensive results, but were unable to fully address the comments of ev39, hF11, and Qz4N. While the authors seem to have addressed comments from B9Fi, after discussion, B9Fi ultimately lowered their score. A consensus was reached that the manuscript is not suitable for acceptance at the current state. While the authors have provided comprehensive experiments, which will improve the overall manuscript, it would require significant the changes in the submission.

---

### Decision · Program_Chairs · 2025-01-22

Reject